# Nursing Profession Self-Efficacy Scale—Version 2: A Stepwise Validation with Three Cross-Sectional Data Collections

**DOI:** 10.3390/healthcare11050754

**Published:** 2023-03-03

**Authors:** Arianna Magon, Gianluca Conte, Federica Dellafiore, Cristina Arrigoni, Irene Baroni, Alice Silvia Brera, Jennifer Avenido, Maddalena De Maria, Alessandro Stievano, Giulia Villa, Rosario Caruso

**Affiliations:** 1Health Professions Research and Development Unit, IRCCS Policlinico San Donato, San Donato Milanese, 20097 Milano, Italy; 2Department of Public Health, Experimental and Forensic Medicine, Section of Hygiene, University of Pavia, 27100 Pavia, Italy; 3Department of Biomedicine and Prevention, University of Rome Tor Vergata, 00133 Rome, Italy; 4Master of Science in Nursing and Midwifery Degree Course, Vita-Salute San Raffaele University, 20132 Milano, Italy; 5Centre of Excellence for Nursing Scholarship, OPI, 00173 Rome, Italy; 6Department of Clinical and Experimental Medicine, University of Messina, 98100 Messina, Italy; 7Center for Nursing Research and Innovation, Vita-Salute San Raffaele University, 20132 Milano, Italy; 8Department of Biomedical Sciences for Health, University of Milan, 20133 Milano, Italy

**Keywords:** care, Mokken scale analysis, nursing, psychometric, scale, self-efficacy, self-report

## Abstract

Background: The nursing professional self-efficacy scale (NPSES) is one of the most used self-reporting tools for assessing nursing self-efficacy. Its psychometric structure was described differently in several national contexts. This study aimed to develop and validate version 2 of the NPSES (NPSES2), which is a brief version of the original scale selecting items that contribute to stably detecting attributes of care delivery and professionalism as descriptors of salient aspects of the nursing profession. Methods: Three different and subsequent cross-sectional data collections were employed to reduce the number of items to generate the NPSES2 and validate its new emerging dimensionality. The first (June 2019–January 2020) involved 550 nurses and was used to reduce the number of the original scale items by using a Mokken scale analysis (MSA) to ensure the selection of items consistently with the invariant item ordering properties. The subsequent data collection was performed to conduct an exploratory factor analysis (EFA) involving 309 nurses (September 2020–January 2021), and the last data collection (*n* = 249) was performed to cross-validate with a confirmatory factor analysis (CFA), the most plausible dimensionality derived from the EFA (June 2021–February 2022). Results: The MSA led to the removal of twelve items and retention of seven items (Hs = 0.407, standard error = 0.023), which showed adequate reliability (rho reliability = 0.817). The EFA showed a two-factor solution as the most plausible structure (factors loading ranged from 0.673 to 0.903; explained variance = 38.2%), which was cross-validated by the CFA that showed adequate fit indices: χ^2^ (13, N = 249) = 44.521, *p* < 0.001; CFI = 0.946; TLI = 0.912; RMSEA = 0.069 (90% CI = 0.048–0.084); SRMR = 0.041. The factors were labeled as care delivery (four items) and professionalism (three items). Conclusions: NPSES2 is recommended to allow researchers and educators to assess nursing self-efficacy and inform interventions and policies.

## 1. Introduction

In the social cognitive theory, self-efficacy was described as individuals’ belief in their ability to succeed in a specific task or accomplish a specific goal [1]. In other words, self-efficacy is the belief in one’s capabilities to organize and execute the courses of action required to manage challenging situations. Self-efficacy is a psychological construct closely related to self-esteem, optimism, and self-mastery, even if it boldly differs from other theoretical concepts because self-efficacy pertains specifically to people’s belief in their ability to achieve goals [2]. In contrast, other concepts (e.g., self-esteem) represent an overall sense of self-worth or positive views regarding aspects of life (e.g., optimism) [2,3].

Over the last two decades, the literature highlighted several reasons why assessing and promoting self-efficacy is significant to several professional groups [4,5,6,7,8]. High self-efficacy is mainly associated with a greater sense of control, motivation, and resilience when facing challenging situations [9]. Individuals with high self-efficacy tend to have greater control over their environment and the challenging situations they encounter because they believe in their ability to influence the outcome of events and take the initiative to make things happen [10]. They also tend to be more motivated to set and achieve goals because they believe in their ability to succeed [11].

In nursing, self-efficacy was extensively studied to determine which factors influence it and which outcomes are influenced by different self-efficacy levels [12]. Nurses with high self-efficacy tend to be more resilient in the face of challenging situations, such as those brought to nurses by the COVID-19 pandemic [13,14]. On the one hand, nurses with high self-efficacy may also cope with stress more effectively, as they believe in their ability to handle stressful situations. On the other hand, nurses with low self-efficacy may experience more stress, burnout, and job dissatisfaction [4,6,8,15,16]. In addition, nurses with low self-efficacy may be less likely to take on leadership roles and may have more difficulty adapting to changes in the healthcare system [7].

Nursing profession self-efficacy is an important aspect of nursing practice that can impact patient outcomes [17,18,19], job satisfaction, level of stress, and overall performance [4,20,21]. Nurses with higher self-efficacy tend to have better coping mechanisms, take on leadership roles [5,6], and provide patient-centered care [7]. To promote self-efficacy in nursing, nurses can engage in continuing education, set achievable goals, seek feedback and support from colleagues and supervisors [8], and work in a positive work culture [15].

For assessing nursing self-efficacy, self-report scales are generally targeted at specific tasks of clinical, managerial, or academic roles [17,18,19]. Even having task-specific scales is important for exploring specific associations between task-specific self-efficacy and its related outcomes, having a broader measurement of nursing profession self-efficacy is important as well when a relationship with broader outcomes (e.g., stress, burnout, and job dissatisfaction) requires being explored [16]. In this regard, the most used self-report scale is the nursing professional self-efficacy scale (NPSES), which was developed in 2016 and translated into several languages (e.g., Korean, Albanian, Turkish) [22,23,24,25].

The NPSES is a 19-item self-report scale that measures self-efficacy in the nursing profession [22,23,24,25]. In its original psychometric structure, NPSES encompassed two specific domains: attributes of caring situations (twelve items) and professionalism situations (seven items) [17]. The original psychometric structure was aligned with the dimensionality of tools aimed at measuring the core nursing professional values [26]. However, further adaptation of the NPSES (e.g., in the Albanian and Korean contexts) showed a slightly different dimensionality from the original one, where four or five dimensions emerged for the psychometric evaluations [22,23,24]. For instance, in the Albanian version, the most plausible psychometric structure was based on “nursing care procedure situations”, “nursing research situations”, “nursing ethics situations”, and “nursing practice situations” [22]. In the Korean version [23], the factors were “professional”, “advocating”, and “caring”, and other versions exist [24]. This multitude of factor structures undermines the possibility of comparing results in cross-national research. The different psychometric characteristics in different countries might be related to a slightly different interpretation of the underlying significance of some items that might reflect context-specific aspects.

More precisely, the differences that emerged in the psychometric structure of the NPSES might be related to the presence of items in the original version that may reflect specific characteristics of the original context where the tool was developed (Italy), such as in relation to “how nurses report to the regulatory authority abuse or unethical behavior of colleagues” in the domain of professionalism. Thus far, the psychometric structure of the NPSES is weak, acknowledging the need for several factor structures to explain its dimensionality in different cultural contexts. A weak factor structure undermines the possibility of comparing results in studies using the same scale in different languages. For this reason, removing items that might contribute to generating different interpretations between national contexts could be an adequate strategy to develop an updated version of the NPSES. Therefore, eliminating items that generate different interpretations in different contexts helps to strengthen the psychometric structure of the NPSES and its capacity to adequately detect nursing self-efficacy. In addition, eliminating ambiguous items develops a short version of the NPSES that might also help educators and researchers measure nursing self-efficacy when measuring multiple theoretical constructs is needed (e.g., studies or educational initiatives including several assessments). Therefore, this study aimed to develop and validate version 2 of the NPSES (NPSES2), which is a brief version of the original scale with select items that contribute to stably detecting care delivery and professionalism as descriptors of salient aspects of the nursing profession.

## 2. Materials and Methods

### 2.1. Design

This study had an observational design with three cross-sectional data collections and it was approved by the authors of the original NPSES. The first data collection was aimed at reducing the number of items of the NPSES by employing a Mokken scale analysis (MSA) focused on identifying items that do not fit well with the underlying construct being measured [27]. Therefore, no changes in the wording of the items were performed, and the NPSES2 was the result of the items retained by the MSA. When NPSES2 was developed, a second data collection was needed for testing, with exploratory factor analysis (EFA), the underlying domains that may represent the most plausible dimensionality of the scale [28]. Finally, the third data collection was aimed at corroborating the dimensionality previously hypothesized by cross-validating it with confirmatory factor analysis (CFA) [29].

### 2.2. Instrument

The NPSES is a self-report scale that assesses nurses’ self-efficacy in their professional practice. The original scale included 19 items that measure care characteristics and professional situations. The respondents rate their level of agreement with each element of the NPSES scale on a 5-point Likert scale, with scores ranging from 1 (completely disagree) to 5 (completely agree). The final score is 0–100 and is standardized following the scoring procedure indicated in the original scale [17], where higher scores indicate higher self-efficacy. Cronbach’s alpha was generally adequate in several previous studies: 0.830 for the original NPSES scale [17], 0.930 for the Korean version [24], and 0.910 for the Albanian version [22]. Its original dimensionality encompassed two factors labeled as attributes of caring situations (twelve items) and professionalism situations (seven items) [17], while four factors were detected in the Albanian version [22] and five factors in the Korean one [24]. Each translated version of the NPSES was cross-culturally adapted in the validation studies [22,23,24,25]. In this study, no cross-cultural adaptation was required because the study was performed in the same language as the original one (Italian).

### 2.3. Samples, Sample Sizes, and Procedures

In a previous simulation study for determining the sample size required for performing MSA [30], six fixed characteristics were specified and similar to those required for performing this analytical procedure for the NPSES. Firstly, the distribution of the latent variables was set to bivariate standard normal. Secondly, two latent variables were used in the analysis. Thirdly, the number of answer categories for each item was set at 5 (NPSES has a 5-point Likert scale as well). Fourthly, the lower bound (c) was set to the default value of 3. Fifthly, 100 replications were conducted for each design cell. Finally, the location parameters of the J items were equidistantly spaced, with the location parameters of each item defined according to its position in the set. This resulted in different sets of location parameters being defined for different items. From that simulation [30], the ideal range of respondents to perform an MSA was between 250 and 600 to ensure per-element accuracy in the analytics.

For this reason, the first data collection (Sample A) involved a hospital in northern Italy (Lombardy) and one in southern Italy (Campania) with roughly 900 eligible nurses to be involved. A preliminary screening performed in the institutional records of the involved hospitals helped researchers identify eligible nurses with characteristics in line with inclusion/exclusion criteria. Eligible nurses had to fulfill inclusion criteria consistent with previous studies aimed at developing self-report measurements [18,21,31,32,33]: full-time work contracts and more than six months of experience in the same context to increase the population homogeneity. The data collection was performed between June 2019 and January 2020 by sending an invitation to eligible nurses to be enrolled in the study. A study information sheet, which was sent to their institutional email, included information regarding the aim, method, and data protection policy. After reading it, nurses willing to be involved had to sign an electronic informed consent form before filling out an online form containing socio-demographic and professional information and the NPSES. The validity of this approach to collecting data was provided by the consistency with previous research and by the analytical method employed in this study [17,18,21,31,32,33]. The following socio-demographic and professional variables were collected: sex (male, female, other), marital status (married, unmarried), education (equal to or higher than a Bachelor of Nursing Science or equivalent titles, higher than bachelor’s degree), clinical area of practice (medical wards, surgical wards, critical area, outpatient services, other), age (years), and work experience (years). The analytics on Sample A to reduce the number of items of the NPSES were performed between February and August 2020.

Given a preliminary exploration of the behaviors of the selected items in Sample A, it was possible to use a Monte Carlo simulation for estimating the sample size required for exploratory structural equation modeling, as it allows researchers to take into account the complexity of the data, the number of factors to be extracted, and the desired level of precision and power. The power in the Monte Carlo simulation was defined as the power needed to reject the null hypothesis that ᴨ (the parameter) = 0, and it was the proportion of significance of the simulation study. Power ≥ 0.80 was desirable. The Monte Carlo simulation was performed in Mplus version 8.1 (Los Angeles, CA: Muthén & Muthén) by employing 1000 replications (seed = 45,335; the residual variances of the factor indicators were 0.36; factor variances were fixed to one; factor correlation set to 0.70). A sample size of 200 was needed for a power of 0.81 or 350 for a power of 0.91 including 15% of missing data under the framework of missing at random (MAR). For this reason, as per Sample A, eligible nurses (*n* = 440) from a hospital in northern Italy (Lombardy) different from the one previously involved, were invited to participate in the second round of data collection with a similar procedure and inclusion criteria to the one described in relation to the first data collection. The data collection was performed between September 2020 and January 2021, while data analysis in Sample B lasted until July 2021.

Considering the factor loadings derived from the EFA performed on Sample B, a second Monte Carlo simulation was used to estimate the sample size required for the CFA in Sample C needed for cross-validating the most plausible factor structure that emerged from Sample B. The model was performed in Mplus version 8.1 (Los Angeles, CA: Muthén & Muthén) by employing 1000 replications (seed = 45,335; the residual variances of the factor indicators were 0.32; factor variances were fixed to one; factor correlation set to 0.72; the mean of factor loading for the first factor was 0.79 and 0.70 for the second factor). A sample size of 140 was needed for a power of 0.81 (without missing data) and 280 for a power of 0.91, including 15% of missing data under the hypothesis of MAR. For this reason, a third hospital in Milan (Lombardy) was involved in the data collection (period: from June 2021 to February 2022) by employing the same procedure and inclusion criteria previously described (eligible nurses = 385).

Overall, the median time to complete the data collection in the three samples was 8 min.

### 2.4. Ethical Statement

This study was conducted following the principles of the Declaration of Helsinki in performing research involving human subjects. The institutional review board of the Italian Association of Cancer Nurses approved the protocol of the study (n.pr/2/2019), and each center provided approval to be involved. The chief nursing officers of each participating hospital received specific education to be responsible for collecting data properly in their settings. Before data collection, participants were informed about the study’s objectives, methods, research design, the confidentiality of the data, and the option to participate.

### 2.5. Statistical Analysis

For each data collection, descriptive statistics were used to summarize the study information. The characteristics of each sample were inferentially compared to ascertain the level of homogeneity related to the socio-demographic and professional variables of the respondents in each sample. The chi-square test (χ^2^) was used to compare qualitative variables, and one-way analysis of variance (ANOVA) was used to compare quantitative variables.

In Sample A, an MSA was performed using the package “mokken” from the statistical program R (R Foundation for Statistical Computing, Vienna, Austria). MSA is a non-parametric method for evaluating the unidimensionality of a set of items in relation to their latent trait (it means “one underlying theoretical construct”; in this case, it is self-efficacy). One of the main advantages of the MSA is that it allows authors to evaluate each item hierarchically and choose the most pertinent ones to evaluate the single latent trait. After having determined with the automated item selection procedure (AISP) that the items of the NPSES referred to a single scale, each item was evaluated in terms of its ability to discriminate between nurses who scored high and low on the scale as a whole. This evaluation was performed by calculating the coefficient of scalability (H) for each item (Loevinger’s coefficient), which ranges from 0 to 1, followed by an assessment of the violations from monotonicity (a maximum of 80 violations are considered acceptable) (coefH function). A value of H close to 1 indicates that the item is highly discriminatory and therefore contributes strongly to the unidimensionality of the scale, while H values equal to or greater than 0.30, 0.40, or 0.50, respectively, showed weak, moderate, or strong scales. The results of the analysis also provided lower and upper bounds for the H values. This approach allows authors to evaluate the items hierarchically and choose the most pertinent ones to evaluate the single latent trait. Loevinger’s coefficients (H values) were calculated for the scale level (Hs), individual items (Hi), and pairs of items HiJ. Items with inadequate H values and/or violations from invariant item ordering (IIO) were deleted from the scale during the IIO procedure performed to reduce the number of items. Molenaar–Sijtsma rho reliability was used to assess the internal consistency.

In Sample B, analytics was performed by using IBM SPSS^®^ Statistics for Windows version 28 (IBM Corp., Armonk, NY, USA). The Kaiser–Meyer–Olkin test (KMO) and Bartlett’s sphericity test were employed to assess that the collected data were appropriate for EFA. EFA was performed from the covariance matrix of the items, extracting factors by considering a previous study [1], the interpretation of the scree plot, and factors with eigenvalues ≥ 1. Factors were rotated using the Promax rotation to obtain an interpretable solution. Parallel analysis with Monte Carlo simulations was used to confirm the number of factors that more likely represent the dimensionality of the NPSES2. McDonald’s ω was used to assess the internal consistency.

In Sample C, the CFA was performed in Mplus version 8.1 (Los Angeles, CA: Muthén & Muthén) to cross-validate the dimensionality explored with EFA. The χ^2^, the ratio between χ^2^ and degrees of freedom, the comparative fit index (CFI), the Tucker–Lewis index (TLI), the Root Mean Square Error of Approximation (RMSEA), and the Standardized Root Mean Square Residual (SRMR) were used to determine if the model fitted to sample statistics. Adequate fit indexes were: CFI ≥ 0.90, TLI ≥ 0.90, RMSEA lower than 0.080, and SRMR lower than 0.1.

The scores of the NPSES2 were 0–100 standardized. All the analyses were performed with two-sided null hypotheses and alpha = 5%. Less than 5% of missingness in the three data collections was managed with an available-case analysis approach.

## 3. Results

### 3.1. Socio-Demographic and Professional Characteristics in the Three Samples

The respondents’ characteristics of each sample are shown in Table 1. In Samples A, B, and C, the respondents were, respectively, 550 (response rate = 61.1%), 309 (response rate = 70.2%), and 249 (response rate = 64.7%). Most of the respondents in the three samples were female nurses ranging from 66.7% (*n* = 367; Sample A) to 75.1% (*n* = 187; Sample C), and females were slightly more represented in Sample C (χ^2^ (1, N = 1108) = 6.59, *p* = 0.037). The majority of nurses were married, with no significant differences between samples (χ^2^ (1, N = 1108) = 2.46, *p* = 0.292). The educational backgrounds showing higher courses than a bachelor’s degree were significantly more frequent in Sample B (33.9%; *n* = 105), followed by Sample A (20.2%, *n* = 111), and Sample C (16.1%; *n* = 40) (χ^2^ (1, N = 1108) = 30.16, *p* < 0.001). Medical wards were the most frequent clinical ward, even if in Sample C, nurses from medical wards were significantly higher than those in Samples A and B (χ^2^ (4, N = 1108) = 140.6, *p* < 0.001). Nurses from Sample B and C had a mean age significantly lower than the mean age of nurses in Sample A (respectively, 38.96 ± 10.55 years, 38.25 ± 10.92 years, and 38.96 ± 10.55 years) (F (2, N = 1107) = 4.62, *p* = 0.010). Likely, nurses from Sample B and C had a mean work experience significantly lower than the mean age of nurses in Sample A (respectively, 14.66 ± 11.23 years, 13.36 ± 13.60 years, and 15.91 ± 10.49 years) (F (2, N = 1108) = 4.42, *p* = 0.012).

### 3.2. MSA in Sample A

The AISP selection did not show items that had to be removed for inconsistencies with the original scale. The assessment of Loevinger’s coefficient of homogeneity of the 19 items showed that Hi values ranged between 0.357 (item 13) to 0.500 (items 3 and 9) (Hs = 0.432, standard error (SE) = 0.020), indicating moderate scalability. The initial assessment of the monotonicity violations showed that all the items reported less than 80 violations, ranging from 0 (item 8) to 69 (items 13 and 17). Therefore, no items were removed from the scale in this stage.

The IIO procedure is shown in Table 2. If a scale is invariant, it means that it is stable and the items are measuring the same underlying construct regardless of their order of presentation; therefore, in this stage, items presenting violations of invariant item ordering were removed, and the scale without these items was re-tested for its invariant item ordering in a stepwise process until no violations were detected. Overall, two steps were necessary. In the first step (*H^T^* = 0.199), items were hierarchically ordered based on their mean scores, and items with significant violations of the invariant item ordering were items 1, 5, 6, 7, 9, 10, 11, 12, 13, 15, 18, and 19. The second step (*H^T^* = 0.301) did not show any violations, and items 2, 3, 4, 8, 14, 16, and 17 represented the Mokken scale (NPSES2). The Loevinger’s coefficient of homogeneity (Hi) ranged between 0.302 (SE = 0.033), which was item 8, and 0.471 (SE = 0.026), which was item 3. At the NPSES2 level, Loevinger’s coefficient of homogeneity (Hs) was 0.407 (SE = 0.023), and the scale showed adequate reliability (Molenaar–Sijtsma rho reliability = 0.817).

### 3.3. EFA in Sample B

The KMO was equal to 0.807, and Bartlett’s test was significant (χ^2^ (21, N = 309) = 897.57 *p* < 0.001), indicating the factorability of the matric of covariances obtained from the responses. The model with a two-factor structure was supported by the analysis of the eigenvalues (also confirmed by a parallel analysis with random data eigenvalues employing a Monte Carlo simulation), the scree plot, and the clarity of the interpretation of the item–factors relationships. Therefore, the two-factor solution was considered the most plausible dimensionality of the NPSES2. Its factor loadings are shown in Table 3 and ranged from 0.673 (item 8) to 0.903 (item 3). The factors were labeled as care delivery (former items 2, 3, 4, and 8; explained variance = 38.2%) and professionalism (former items 14, 16, and 17; explained variance = 29.02%). McDonald’s ω values for care delivery and professionalism were, respectively, 0.842 and 0.815.

### 3.4. CFA in Sample C

The posited model with two factors showed adequate fit to sample statistics: χ^2^ (13, N = 249) = 44.521, *p* < 0.001; χ^2^/DF = 3.4; CFI = 0.946; TLI = 0.912; RMSEA = 0.069 (90% CI = 0.048–0.084); SRMR = 0.041. The factor loadings are shown in Table 3 and ranged from 0.614 (item 17) to 0.889 (item 3). Residual variances ranged from 0.210 (SE = 0.045; item 2) to 0.523 (SE = 0.069; item 7). McDonald’s ω values for care delivery and professionalism were, respectively, 0.861 and 0.821. The two factors were positively correlated (r = 0.776; *p* < 0.001). No alternative models were necessary to be tested for cross-validating the dimensionality of the NPSES2 derived from the EFA.

### 3.5. NPSES2 Scores in the Three Samples

Figure 1 shows the distribution in each sample of self-efficacy toward care delivery and professionalism. Overall, care delivery showed higher scores than professionalism in each sample (*p* < 0.01); its mean scores in Samples A, B, and C were, respectively, 57.65 ± 11.93, 54.41 ± 12.50, and 58.54 ± 12.76. The mean score of professionalism in Samples A, B, and C were, respectively, 47.56 ± 15.63, 47.18 ± 16.91, and 48.18 ± 15.98.

## 4. Discussion

This study showed that the NPSES2 is a brief, valid, and reliable self-report measure for assessing nursing profession self-efficacy. The previous tool was based on 19 items [17] with a dimensionality described to be context-specific in several studies [22,23,24,25]. More precisely, the original dimensionality was based on a two-factor structure, which was not confirmed when the NPSES was used in international contexts [22,23,24,25]. For this reason, the results derived from this study that removed ambiguous items are important for at least five reasons that can be summarized in aspects related to validity, reliability, significance, research reasons, and practicality.

This study improved the validity of the scale compared to the previous version. A valid self-efficacy scale is essential to ensure that it measures what it intends to measure, leading to more accurate conclusions and a better understanding of the construct. The deleted items from the NPSES to develop the NPSES2 were those violating the invariant item ordering. The invariant item ordering is pivotal because it ensures that the factor structure that is obtained is truly representative of the underlying construct being measured rather than being an artifact of the order in which the items were presented [34]. No evidence of invariant item ordering was previously described in relation to the NPSES. This approach in developing the NPSES2 ensured that the factor structure was easily interpretable, meaning that the factors that were extracted in the subsequent EFA were meaningful and represented the underlying constructs being measured. In addition, the MSA and the item selection based on removing those violating the invariant item ordering ensured higher stability than the previous version [35] and more generalizability because evidence of invariant item ordering is associated with high external validity [36].

The employed approach, based on a stepwise method of three separate data collections to refine the scale from the item selection procedure (MSA in Sample A) to the cross validation of the dimensionality (CFA in Sample C), also provides increased reliability of the NPSES2 compared to the previous version. While invariant item ordering refers to the property of a scale that the order of items does not affect the factor structure that is extracted, reliability refers to the consistency of the results of a scale; however, the two properties are associated because the invariant item ordering procedure ensured that the factor structure was consistent, regardless of the order in which the items are presented. This consistency is important for the scale’s reliability, as it ensures that the factors extracted are accurate and not affected by the specific order of the items. In addition, the MSA ensured that the factor structure could be easily replicable, meaning that the factors that were extracted could be replicated in different samples or settings, overcoming the issues detected by previous studies [22,24]. This replicability is also important for the scale’s reliability because it ensures that other researchers can replicate the factors extracted and that the results may be more generalizable than the ones obtained with the previous version of the scale.

Self-efficacy is a cognitive-motivational construct that has been linked to a wide range of outcomes, such as academic and career success, mental and physical health, and well-being [2,3]. In the NPSES2, the invariant item ordering ensured by the MSA provides increased confidence in the conclusions and findings of studies that will use this tool, which means that the significance of measuring self-efficacy by using NPSES2 has been improved compared with the significance that a scale with uncertain dimensionality might have in research settings. In other words, the improved validity and reliability characteristics of the NPSES2 compared to the former version might also contribute to sustaining the significance of assessing nursing profession self-efficacy.

In this study, the general levels of nursing profession self-efficacy in the three samples seemed to be generally limited (see Figure 1). However, it is impossible in this study to determine which score can discriminate a situation of inadequate self-efficacy that could be associated with worsened outcomes (e.g., lower work performance, lower well-being) because further research is needed. Future studies should determine cutoff scores of the NPSES2 by using two possible approaches: the Receiver Operating Characteristic (ROC) curve or a criterion-related cutoff identification. The first method involves plotting the true positive rate (sensitivity) against the false positive rate (1-specificity) for different cutoff scores in relation to a dichotomic outcome (e.g., intention to leave the profession) [37]. The optimal cutoff score is the one that maximizes the area under the ROC curve (AUC). The second possible method involves determining cutoff scores based on their correlation with an external criterion, such as a behavioral assessment. This method is useful when the scale is used to predict a specific outcome, such as a behavior.

### 4.1. Limitations and Strengths

This study has several limitations that are required to be acknowledged. First, the cross-sectional nature of the data collection rounds did not allow researchers to take into account the relationship between factor structure and time. Second, cross-sectional studies are susceptible to selection bias, meaning that the sample of participants may not be fully representative of the population being studied. In fact, in this study, we detected small variations in relation to age, sex, and work context. In general, the characteristics of the respondents are similar to the ones reported for the general population of Italian nurses [38]. Third, no information is thus far available in relation to the cutoff scores that may be used to detect inadequate self-efficacy. This aspect limits the current interpretation of the scores derived from the NPSES2. Fourth, the limited number of institutions involved in the study (four hospitals) might limit the generalizability, even if the sample size for each analysis was adequate. In this regard, generalizability might also be undermined by the differences in the characteristics of the respondents in the three samples, even if these differences might reflect “real-life” differences between hospitals. Lastly, it was not feasible to test the multi-group invariance between the three samples to avoid performing multiple and different psychometric tests on the same sample because it can lead to drawbacks with the validity of the results.

This study has several strengths and innovations. Using multiple data collection rounds in different samples analyzed with different analytic methods (MSA, EFA, and CFA) increases the rigor of the study and provides a more comprehensive assessment of the psychometric properties of the instrument. This approach allows for a more accurate and reliable evaluation of the reliability and validity of the NPSES2. In addition, the fewer items in the NPSES2, which is a short form of the NPSES, can be useful in various settings, including academic and clinical contexts, and can help reduce respondent burden without sacrificing the measurement quality. Finally, a clearer psychometric structure derived from the several analytical steps can improve its utility and validity by providing a more consistent and coherent assessment of nursing professional self-efficacy. The clearer structure can also improve the interpretability of the scores and help identify areas where nurses may need further development or support.

### 4.2. Study Implications

This study produces a brief, valid, and reliable self-report scale of self-efficacy, allowing researchers and educators to make accurate and reliable conclusions about the construct of self-efficacy in a specific population, which can improve the understanding of the construct and help to inform interventions and policies. In relation to the practicality and research reasons, a brief self-report scale such as the NPSES2 is more practical to use in research and practice because it is less time-consuming for participants to complete and easier for researchers to administer [39]. Therefore, NPSES2 utilization could increase the response rate and reduce participant burden in research initiatives. This aspect is particularly meaningful concerning the need for several measurements when self-efficacy is studied with its antecedents and outcomes. This hypothesis is rooted in the findings of a previously performed meta-analysis showing a strong correlation between response rate and questionnaire length, where response rates were lower for longer scales or questionnaires [39]. For these reasons, we recommend using NPSES2 when it is needed to assess professional self-efficacy among nurses, and cross-national validation studies of NPSES2 are required before using it in contexts different from the one where it was developed.

## 5. Conclusions

This study developed and validated the NPSES2, a seven-item self-report scale measuring two aspects of nursing profession self-efficacy: care delivery (four items) and professionalism (three items). Future research should determine which score of NPSES2 can identify risks for nurses or inadequate areas of self-efficacy with a likelihood of being associated with negative professional or health-related outcomes. It is recommended to test the NPSES2 validity and reliability in different contexts from the one where the scale was developed (Italy), and the wording of the items required to be culturally adapted in target languages different than Italian.

## Figures and Tables

**Figure 1 healthcare-11-00754-f001:**
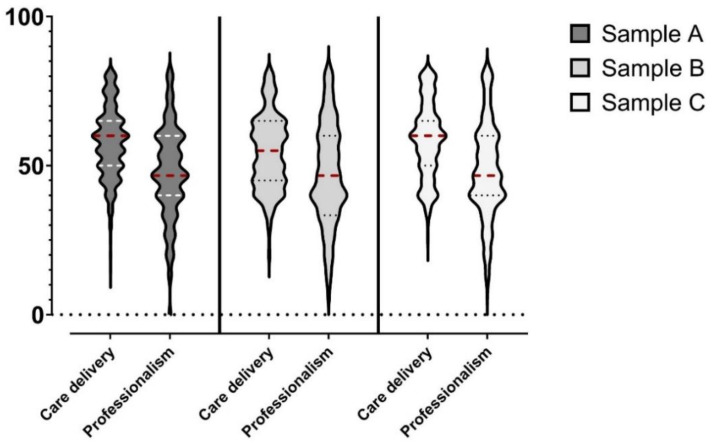
Violin plots describing the computed scores of the NPSES2 in each sample.

**Table 1 healthcare-11-00754-t001:** Characteristics of the respondents in the three samples (N overall = 1108).

		Sample A (*n* = 550)		Sample B (*n* = 309)		Sample C (*n* = 249)		Comparisons
Variables		N	%	N	%	N	%	
Sex								
	Female	367	66.7	223	72.3	187	75.1	χ^2^ (1, N = 1108) = 6.59, *p* = 0.037
Marital status								
	Married	315	57.3	167	54	151	60.6	χ^2^ (1, N = 1108) = 2.46, *p* = 0.292
Education								
	Higher than a bachelor’s degree	111	20.2	105	33.9	40	16.1	χ^2^ (1, N = 1108) = 30.16, *p* < 0.001
Clinical area of practice								
	Medical wards	150	27.3	120	38.8	174	69.9	χ^2^ (4, N = 1108) = 140.6, *p* < 0.001
	Surgical wards	129	23.5	71	23.0	24	9.6	
	Critical care units	134	24.4	50	16.2	12	4.8	
	Outpatient services	54	9.8	31	10.0	15	6.0	
	Other	83	15.1	37	12.0	24	9.6	
Age								
	Years (mean; SD)	40.46	9.9	38.96	10.55	38.25	10.92	F (2, N = 1107) = 4.62, *p* = 0.010
Work experience								
	Years (mean; SD)	15.91	10.49	14.66	11.23	13.36	13.6	F (2, N = 1108) = 4.42, *p* = 0.012

Legend: SD = standard deviation.

**Table 2 healthcare-11-00754-t002:** Round of IIO and Htrans (HT) per each step and Mokken scale (N = 550).

Item	Label	Mean	SD	Step 1	Step 2	Hi	(SE)	Hs (SE)	Rho Reliability
**Item 8**	Promote patient confidentiality and privacy	4.23	0.84	0	0	0.302	(0.033)	0.407 (0.023)	0.817
Item 15	Safeguard the right of patients’ privacy and confidentiality in processing health-related data	4.15	0.83	3	removed				
Item 10	Use the support of other colleagues to evaluate a particular situation or problem	4.06	0.77	1	removed				
Item 12	Safeguard the legal and moral rights of patients	3.98	0.82	2	removed				
Item 5	Deliver individualized healthcare based on the principle of equity without discrimination	3.96	0.77	2	removed				
Item 9	Examine the quality (accuracy/completeness) of clinical documentation	3.93	0.78	1	removed				
Item 19	Practicing nursing, recognizing and addressing the ethical/moral dilemmas	3.89	0.82	6	removed				
**Item 3**	Safeguard health and safety at the community and social levels	3.83	0.76	0	0	0.471	(0.026)		
**Item 4**	Ensure healthcare is always delivered in line with the highest professional standards	3.79	0.74	0	0	0.456	(0.029)		
Item 1	Respect patients and their autonomy (e.g., principles of freedom of choice or self-determination)	3.79	0.83	2	removed				
Item 18	Ensure a fair use of the resources that I have in my professional practice	3.74	0.81	1	removed				
Item 6	Compensate for the inefficiencies that may occur in the facility where I work	3.69	0.77	1	removed				
**Item 2**	Practice following evidence-based nursing to deliver safe and effective care	3.68	0.76	0	0	0.450	(0.026)		
Item 11	Implement the results of research in professional practice	3.61	0.82	1	removed				
Item 13	Refuse to participate in treatments if it is contrary to the shared professional values (e.g., national code of conduct)	3.57	0.98	1	removed				
**Item 14**	Take part in nursing research	3.49	1.00	0	0	0.386	(0.031)		
Item 7	Promote the use of ethics consultation for ethical dilemmas related to caring work	3.49	0.86	1	removed				
**Item 16**	Collaborate with nursing associations and professional representatives to ensure the best standards of care in my practice	3.4	0.92	0	0	0.467	(0.026)		
**Item 17**	Report any abuse or unethical behavior of colleagues to the appropriate Regulatory Authority	3.24	1.01	0	0	0.341	(0.033)		
*H^T^*				0.199	0.301				

Legend: IIO = invariant item ordering procedure; SD = standard deviation; Hi = Loevinger’s coefficient of homogeneity (Item level); Hs = Loevinger’s coefficient of homogeneity (Scale level); SE = Standard Error, *H^T^* = Htrans is a value representing the accuracy of the IIO procedure (values higher than 0.30 are considered accurate); items indicated in bold are retained in the Mokken scale.

**Table 3 healthcare-11-00754-t003:** Factor loadings of the EFA performed in Sample B and CFA conducted in Sample C.

	Sample B (*n* = 309)		Sample C (*n* = 249)	
	Care Delivery	Professionalism	Care Delivery	Professionalism
Item 2	**0.845**	0.031	**0.817**	-
Item 3	**0.903**	−0.005	**0.889**	-
Item 4	**0.760**	0.116	**0.809**	-
Item 8	**0.673**	−0.132	**0.616**	-
Item 14	0.148	**0.701**	-	**0.689**
Item 16	−0.04	**0.895**	-	**0.812**
Item 17	−0.102	**0.889**	-	**0.614**
McDonald’s ω	0.842	0.815	0.861	0.821
Explained variance (%)	38.2	29.02	-	-

Note: bold values represent the items kept by an underlying factor with loading > 0.32.

## Data Availability

Raw data are available from the corresponding author upon reasonable request.

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
