# Peer review of "Nursing Profession Self-Efficacy Scale—Version 2: A Stepwise Validation with Three Cross-Sectional Data Collections"

_healthcare, 2023, doi:10.3390/healthcare11050754_

Round 1
Reviewer 1 Report
Please see the attachment

Author Response
Please, find here attached our point-by-point response.

Reviewer 2 Report
ID: healthcare-2212960
Title: Nursing profession self-efficacy scale – version 2: a stepwise validation with three cross-sectional data collections
Thank you for providing a chance to review this manuscript.
Comment: Major Revision.
Detailed information:
Abstract
Methods:
1) Please briefly complement the process of establishing the version 2 of NPSES;
2) The number of the last data collection should be listed.
Results: Important statistical results of EFA and CFA need to be described in more detail.
Introduction
Paragraph 1-2, Page 1-2: These two paragraphs mainly describe the self-efficacy of nurses, but some sentences are repetitive. It is recommended that the authors concise the description.
Paragraph 3, Page 2: This paragraph again describes the definition of self-efficacy, which is repeated in the previous two paragraphs. In addition, I think the content of this paragraph is more appropriate in the first paragraph. Please review the logic of the article.
Paragraph 6, Page 2: What is the psychometric significance of various versions of NPSES?
Paragraph 7, Page 3: Why the NPSES Scale should be modified? More evidence needs to be added.
Materials and Methods
Instrument:
1) Is the use of NPSES approved by the original author?
2) Does NPSES undergo cross-cultural adaptation? If so, show it in the method.
3) Please explain the update process for NPSES version 2 in detail.
Samples, sample sizes, and procedures
1) “As determined in previous simulation research [33], the ideal range of respondents to perform an MSA was between 250 and 600 respondents” More authoritative explanations are needed to prove the sample size of this study is sufficient.
2) Is it representative that nurses come from only two institutions?
3) What are the inclusion and exclusion criteria for participants?
4) Online survey is indeed very convenient, but how to ensure the validity of questionnaire answers?
5) Do samples A, B, and C overlap?
Results
Socio-demographic and professional characteristics in the three samples:
According to the results of this part, the general characteristics of samples A, B and C differ greatly. Are the inclusion and exclusion criteria appropriate? Please explain.
NPSES2 scores in the three samples:
“Overall, care delivery showed higher scores than professionalism in each sample” Is there a statistical difference?
Overall: Whether the authors have further analysis to prove the effectiveness of the NPSES – version 2, such as correlation analysis, ROC curve analysis, etc. In addition, this study is supposed to conduct longitudinal analysis and calculate internal consistency, which can make the results of this study more persuasive.
Discussion
1) It is recommended to describe the results of the original study in the discussion and then compare them with the results of the updated version.
2) Are there any similar studies to adapt NPSES, and what are the conclusions? It can be discussed from this perspective.
3) The authors presented the limitations of this study but ignored the innovation, which needs to be clarified in the paper.
This study explored the validation of t Nursing profession self-efficacy scale – version 2. First, the updating process of the NPSES questionnaire needs to be described in detail in the method, including cross-cultural adaptation and deletion of items. In addition, the authors are expected to pay special attention to the logic of expression, especially the introduction and discussion. Moreover, there are many basic errors in this article, such as grammar, punctuation and so on. Last but not least, finding a native English speaker to improve the writing can considerably improve the quality.
Thank you and my best,
Your reviewer
Author Response

(The authors gave the same response as above.)

Reviewer 3 Report
The peer-reviewed manuscript aims to develop and validate version 2 of the NPSES (NPSES2), which is a shortened version of the original scale, by selecting items that contribute to the stable detection of care delivery and professionalism attributes as descriptors of the salient aspects of the nursing profession.
I believe that the study was conducted reliably, the authors accurately describe all stages of the study. The research groups were selected properly, the conclusions are logical and correspond to the purpose of the work. Congratulations to the authors.
Author Response

(The authors gave the same response as above.)

Reviewer 4 Report
My comments are attached...Great work!

Author Response
Please, find attached our point-by-point response.

Round 2
Reviewer 2 Report
ID: healthcare-2212960
Title: Nursing profession self-efficacy scale – version 2: a stepwise validation with three cross-sectional data collections
Thank you for providing a chance to review this manuscript.
Comment: Minor Revision.
Detailed information:
Materials and Methods
1) Since NPSES version 2 has deleted items in the original questionnaire, will it be understood that it is a shortened version of NPSES?
2) Is it appropriate to further explain Mokken Scale Analysis (MSA) in the introduction or in the method? I believe that many readers do not know the significance of this method.
Results
With respect to the differences between the general characteristics of the three groups of people, I have to question whether the measurement invariance of this version is reliable.
Overall: “Whether the authors have further analysis to prove the effectiveness of the NPSES – version 2, such as correlation analysis, ROC curve analysis, etc. In addition, this study is supposed to conduct longitudinal analysis and calculate internal consistency, which can make the results of this study more persuasive”. On this question, is it achievable?
Thank you and my best,
Your reviewer
Author Response
Dear reviewer, please find our response in the attached file.
